# Mapping global acceptance and uptake of COVID-19 vaccination: A systematic review and meta-analysis

Qian Wang[1,4], Simeng Hu[1,4], Fanxing Du[1], Shujie Zang[1], Yuting Xing[1], Zhiqiang Qu[1], Xu Zhang[1], Leesa Lin[2,3] & Zhiyuan Hou [1✉]

## Abstract

**Background** The COVID-19 pandemic exit strategies depend on widespread acceptance of COVID-19 vaccines. We aim to estimate the global acceptance and uptake of COVID-19 vaccination, and their variations across populations, countries, time, and sociodemographic subgroups.

**Methods** We searched four peer-reviewed databases (PubMed, EMBASE, Web of Science, and EBSCO) for papers published in English from December 1, 2019 to February 27, 2022. This review included original survey studies which investigated acceptance or uptake of COVID-19 vaccination, and study quality was assessed using the Appraisal tool for Cross-Sectional Studies. We reported the pooled acceptance or uptake rates and 95% confidence interval (CI) using meta-analysis with a random-effects model.

**Results** Among 15690 identified studies, 519 articles with 7,990,117 participants are eligible for meta-analysis. The global acceptance and uptake rate of COVID-19 vaccination are 67.8% (95% CI: 67.1–68.6) and 42.3% (95% CI: 38.2–46.5), respectively. Among all population groups, pregnant/breastfeeding women have the lowest acceptance (54.0%, 46.3–61.7) and uptake rates (7.3%, 1.7–12.8). The acceptance rate varies across countries, ranging from 35.9% (34.3–37.5) to 86.9% (81.4–92.5) for adults, and the lowest acceptance is found in Russia, Ghana, Jordan, Lebanon, and Syria (below 50%). The acceptance rate declines globally in 2020, then recovers from December 2020 to June 2021, and further drops in late 2021. Females, those aged < 60 years old, Black individuals, those with lower education or income have the lower acceptance than their counterparts. There are large gaps (around 20%) between acceptance and uptake rates for populations with low education or income.

**Conclusion** COVID-19 vaccine acceptance needs to be improved globally. Continuous vaccine acceptance monitoring is necessary to inform public health decision making.

### Plain language summary

Our ability to control COVID-19 and mitigate its serious effects on our health has largely depended on peoples' willingness to have a COVID-19 vaccine, known as vaccine acceptance. Vaccine acceptance is a complex issue and levels of acceptance vary between different countries and populations. Here, we conduct a systematic search for studies on the topic of COVID-19 vaccine acceptance and analyse their results. We find that the global acceptance rate of COVID-19 vaccines is lower than 70%, with large variations between countries. The lowest acceptance rates are found in Russia, Ghana, Jordan, Lebanon, and Syria. Vulnerable populations with low acceptance rates include pregnant or breastfeeding women, Black people, and those with low socioeconomic status. Our findings highlight differences in vaccine acceptance between different populations, and suggest the need to carefully monitor and improve vaccine acceptance rates.

[1] School of Public Health, Global Health Institute, Fudan University, Shanghai, China. [2] Department of Infectious Disease Epidemiology, London School of Hygiene & Tropical Medicine, London, UK. [3] Laboratory of Data Discovery for Health (D24H), Hong Kong Science Park, Hong Kong SAR, China. [4] These authors contributed equally: Qian Wang, Simeng Hu. ✉email: zyhou@fudan.edu.cn

The COVID-19 pandemic has become the most threatening global health issue[1]. The cataclysmic impact of the COVID-19 pandemic has contributed to an unprecedented pace in COVID-19 vaccine development[2]. Effective vaccine development usually takes almost 10 years, but COVID-19 vaccines have been developed and issued for use within a one-year timeframe[3]. Considering the delta variant, around 85% of the population should get immunity through natural infection or vaccination[4]. Given the powerful capability of the omicron variant to escape neutralizing antibodies elicited by current vaccines, more than 85% of the population need to get immunity[5].

Public confidence and acceptance of COVID-19 vaccines need to be ensured to achieve high vaccination uptake and herd immunity[6,7]. However, the accelerated development and issue process of COVID-19 vaccines may exacerbate public concerns regarding their safety and effectiveness[3]. The novelty of the COVID-19 disease, the anti-vaccine movement, and politicization of the COVID-19 vaccine may also negatively influence vaccine acceptance[8]. Previous studies have investigated public acceptance of COVID-19 vaccines, with substantial heterogeneity across the world[9–13]. Vaccine acceptance is defined as the individual or group decision to accept or refuse, when presented with an opportunity to vaccinate[14]. It is a complex and context specific issue that varies across time, place, and vaccines[15]. With the evolution of the pandemic and widespread dissemination of COVID-19 related misinformation[16], public acceptance may change over time. Although a growing body of literature has investigated public acceptance of COVID-19 vaccination, few studies have systematically reviewed and synthesized the current evidence[3,17–19].

We conducted a systematic review and meta-analysis to estimate the global acceptance and uptake of COVID-19 vaccination, including 1) global acceptance and uptake of COVID-19 vaccination in each population group, 2) cross-country comparison and time trends of vaccination acceptance, and 3) variations in vaccination acceptance and uptake across subgroups according to sociodemographic characteristics. We find that the global acceptance rate of COVID-19 vaccines is lower than 70%, with large variations between countries. The lowest acceptance rates are found in Russia, Ghana, Jordan, Lebanon, and Syria. The acceptance rates decline globally in 2020, then recover in the first half of 2021, and further drop in late 2021. Vulnerable populations with low acceptance rates include pregnant or breastfeeding women, Black people, and those with low socioeconomic status. Our findings highlight differences in vaccine acceptance between different populations, and suggest the need to carefully monitor and improve vaccine acceptance rates.

## Method

**Search strategy and selection criteria**. This review was developed according to the Preferred Reporting Items for Systematic Reviews and Meta-Analyses (PRISMA) guidelines[20]. We employed the following search terms on four peer-reviewed databases (PubMed, EMBASE, Web of Science, and EBSCO): coronavirus terms ("coronavirus disease" OR coronavirus OR coronaviruses OR 2019-nCoV OR 2019ncov OR COVID-19 OR "severe acute respiratory syndrome coronavirus 2" OR SARS-2 OR SARS-COV-2) AND vaccine terms (vaccin* OR immunis* OR immuniz*) AND survey terms (survey OR questionnaire OR poll). All papers published in English from December 1, 2019 to February 27, 2022 were collected with the above search terms in all fields of studies. The detailed search strategy for each database is included in Supplementary Data 1.

Articles were included in this review if they investigated acceptance, willingness, intention, or uptake of COVID-19 vaccination, and if they were original survey studies. We excluded studies that investigated (1) non-COVID-19, clinical-trial, emergency or boosting vaccination acceptance, (2) studies which assessed willingness-to-pay or conditional acceptance, (3) studies without outcomes of interest, or (4) studies that applied continuous variables to evaluate vaccination acceptance. Studies using continuous variables usually adopted different response ranges with no consistent meanings for response options across studies, and there were also no clear cut-off points for vaccination acceptance or refusal in continuous variables. Therefore, studies with continuous variables were excluded in our review for conducting the meta-analysis of vaccination acceptance rate. The following study designs were also excluded: editorials, letters, commentaries, correspondences, study protocols, reviews, qualitative studies, intervention studies, and non-survey studies such as crawling information from social media. Two independent researchers (SH, QW) first screened titles and abstracts, and then scrutinized the full texts to estimate their eligibility. When they had disagreements on study selection, the third researcher (FD) was consulted. The review protocol is available on International prospective register of systematic reviews (PROSPERO) (ID: CRD42021261022). This study was exempt from ethical review due to use of publicly available data.

**Data abstraction and quality assessment**. Two researchers (SH, QW) independently performed article extraction and assessed the quality of included studies. When inconsistency arose, they were asked to discuss and revisit the article until reaching a consensus. We extracted the following information from the included articles: title, first author, publication date, journal, study design, sampling method, sample size, survey period, survey location, target population, and measurement questions about COVID-19 vaccination acceptance. To achieve the study objectives, we also extracted four outcomes: (1) overall acceptance of COVID-19 vaccination (total sample); (2) subgroups' acceptance of COVID-19 vaccination (by gender, age, race, education, and income); (3) overall uptake of COVID-19 vaccination (total sample); and (4) subgroups' uptake of COVID-19 vaccination (by gender, age, race, education, and income). For each included study, we described its characteristics, study design, and primary outcomes in Sheet 1 in Supplementary Data 2.

The Appraisal tool for Cross-Sectional Studies (AXIS tool), a novel critical appraisal tool that addresses study design and reporting quality as well as the risk of bias in cross-sectional studies, was used to assess the quality of the included studies[21]. This tool, shown in Supplementary Data 3, includes three domains and twenty items with a total possible score of 20: quality of reporting (7 items), study design quality (7 items), and the possible introduction of biases (6 items). Given 12 of 20 scores (60%) are considered pass, studies > 12 scores were considered with the high-quality, and data from those high-quality studies were extracted for the review and meta-analysis. The quality assessment scores of each included study are shown in Supplementary Data 4.

**Statistical analysis**. Data organization and meta-analysis were carried out using Microsoft Excel and STATA 15.1 software respectively. Figures were done with R (version 4.1). Acceptance, willingness, or intention of COVID-19 vaccination were categorized into three groups: (1) Yes/ Definitely/ Probably; (2) Unsure/ Neutral/ I don't know; and (3) No/ Definitely not / Probably not (Sheet 1 in Supplementary Data 2). The first one was labelled as "accept", and the latter two were labelled as "vaccine hesitant". For studies covering the vaccinated individuals, we took vaccinated individuals as the "accept" group when

calculating vaccine acceptance rates. The acceptance rate of COVID-19 vaccination was defined as the proportion of participants willing to / accept / intend to / will get vaccination against COVID-19 in total surveyed participants. Uptake status of COVID-19 vaccination were categorized into two groups: (1) Yes and (2) No.

We reported the pooled acceptance or uptake rates and 95% confidence interval (CI). We employed a DerSimonian and Laird random-effects models[22] to conservatively estimate the pooled acceptance or uptake of COVID-19 vaccination, in case of significant heterogeneity ($I^2 > 50\%$) between studies. Variability between studies was determined by the heterogeneity tests with Higgins' $I^2$ statistic. Stratified subgroup analyses were conducted to explore the sources of heterogeneity. A value of $P < 0.05$ was considered statistically significant.

The pooled acceptance or uptake rates of COVID-19 vaccination were estimated by different populations, countries, survey times, and participants' characteristics. We categorized all study participants into seven groups: (1) adults, (2) healthcare workers, (3) patients with chronic diseases, (4) pregnant or breastfeeding women, (5) university students, (6) children and adolescents, and (7) other populations (Sheet 1 in Supplementary Data 2). Other populations were defined as study populations that cannot be categorized into the first six study populations, such as the homeless, those in a particular occupation, and elderly persons with Medicare. We first estimated the pooled acceptance or uptake of COVID-19 vaccination for each population group, and within each population group, we then estimated the acceptance or uptake of COVID-19 vaccination in individual countries by synthesizing all studies from the same country. To compare the trends of COVID-19 vaccination acceptance over time, we reported the pooled acceptance rate of all studies from the same survey month, and developed graphs to illustrate the time trends. For acceptance or uptake estimates from adults, we also conducted subgroup analyses based on their sociodemographic characteristics such as gender, age group, race, education, and income. Additionally, we did a sensitivity analysis of the pooled acceptance rate of COVID-19 vaccination with studies whose sample size was more than 300 (Supplementary Table 1 in Supplementary Information).

**Reporting summary.** Further information on research design is available in the Nature Research Reporting Summary linked to this article.

## Results

A total of 15690 articles were identified through database searches. After excluding 8073 duplicates and a further 6124 from the screening of titles and abstracts, 1493 articles were full-text screened for eligibility (Fig. 1). Ultimately 519 articles were eligible for inclusion, with 7,990,117 participants included in the meta-analysis (Supplementary Data 5). A total of 678 samples were included, as one study may contain several samples across different countries, survey times, and populations. Of 678 samples included, the majority were from the United States ($n = 133$), mainland China ($n = 59$), the United Kingdom ($n = 24$), Saudi Arabia ($n = 40$), India ($n = 21$), Italy ($n = 20$), Turkey ($n = 18$), Canada ($n = 17$), Australia ($n = 16$), and Bangladesh ($n = 13$). The survey time of the included studies covered January 2020 to December 2021. About two thirds of studies (46.2%, 240/519) described results of COVID-19 vaccination acceptance from convenience samples, while 110 studies (21.2%) used purposive sampling to select study participants.

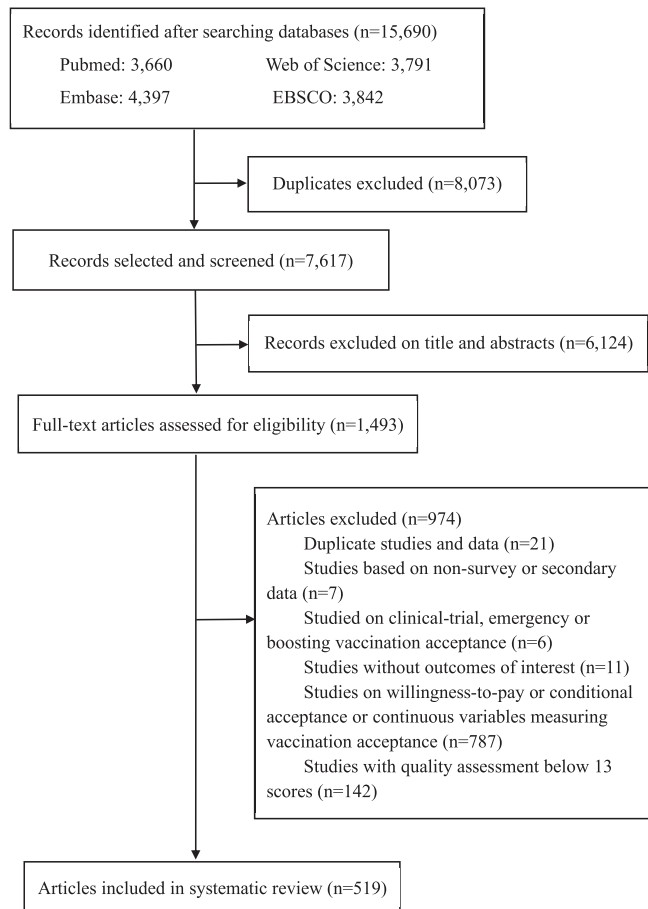

Records identified after searching databases (n=15,690)
  Pubmed: 3,660    Web of Science: 3,791
  Embase: 4,397    EBSCO: 3,842

Duplicates excluded (n=8,073)

Records selected and screened (n=7,617)

Records excluded on title and abstracts (n=6,124)

Full-text articles assessed for eligibility (n=1,493)

Articles excluded (n=974)
  Duplicate studies and data (n=21)
  Studies based on non-survey or secondary data (n=7)
  Studied on clinical-trial, emergency or boosting vaccination acceptance (n=6)
  Studies without outcomes of interest (n=11)
  Studies on willingness-to-pay or conditional acceptance or continuous variables measuring vaccination acceptance (n=787)
  Studies with quality assessment below 13 scores (n=142)

Articles included in systematic review (n=519)

**Fig. 1 Flowchart of study selection.** Selection of original survey studies on acceptance, willingness, intention, or uptake of COVID-19 vaccination is shown via flowchart.

**Global acceptance and uptake of COVID-19 vaccination.** Among the 519 studies with a score > 12, 476 studies investigated the willingness of COVID-19 vaccination, and reported that the pooled acceptance rate of COVID-19 vaccination was 67.8% (95% CI: 67.1–68.6) (Table 1). Acceptance rates varied among different populations; children and adolescents (70.7%, 67.6–73.9) had the highest acceptance rate, followed by adults (69.1%, 68.2–70.1), university students (67.7%, 62.7–72.8), healthcare workers (67.5%, 64.4–70.6), and patients with chronic disease (67.4%, 63.9–70.9), while pregnant/breastfeeding women (54.0%, 46.3–61.7) had the lowest acceptance rate.

404 of 519 studies investigated the proportion of participants who were unwilling to get a COVID-19 vaccine, and found the pooled level to be 20.4% (95% CI:19.6–21.3), as shown in Table 1. The proportion of participants who were unwilling to get a COVID-19 vaccine ranked from high to low for pregnant/breastfeeding women (41.9%, 33.0–50.8), university students (20.7%, 17.2–24.2), adults (19.8%, 18.5–21.1), children and adolescents (19.8%, 17.9–21.7), healthcare workers (19.8%, 17.7–21.9), and patients with chronic diseases (16.9%, 13.9–19.9).

139 of 519 studies investigated the uptake of COVID-19 vaccination, and reported that the pooled uptake rate was 42.3% (95% CI: 38.2–46.5) until November, 2021 (Table 1). Healthcare workers (54.1%, 46.5–61.7) had the highest uptake rate, followed by university students (43.7%, 31.2–56.1), adults (39.7%, 32.4–47.1), patients with chronic disease (39.3%, 31.9–46.7), and children and adolescents (37.9%, 22.5–53.4), while pregnant/breastfeeding women (7.3%, 1.7–12.8) had the lowest uptake rate.

**Table 1 Estimated acceptance and uptake of COVID-19 vaccination by study populations**

| Population groups | Acceptance of COVID-19 vaccines | | | | | | Unwilling to get a COVID-19 vaccine | | | Uptake of COVID-19 vaccines | | |
| --- | --- | --- | --- | --- | --- | --- | --- | --- | --- | --- | --- | --- |
| | Willing to get a COVID-19 vaccine | | | Unsure | | | | | | | | |
| | No. of studies | No. of participants | Estimated acceptance (%, 95% CI) | No. of studies | No. of participants | Estimated unsure rate (%, 95% CI) | No. of studies | No. of participants | Estimated unwillingness (%, 95% CI) | No. of studies | No. of participants | Estimated uptake (%, 95% CI) |
| Overall | 476 | 7967690 | 67.8 (67.1–68.6) | 221 | 1392225 | 20.3 (19.3–21.3) | 404 | 7748451 | 20.4 (19.6–21.3) | 139 | 6179852 | 42.3 (38.2–46.5) |
| Adults | 202 | 7068345 | 69.1 (68.2–70.1) | 88 | 972239 | 19.4 (18.1–20.8) | 177 | 6948770 | 19.8 (18.5–21.1) | 49 | 5877618 | 39.7 (32.4–47.1) |
| Healthcare workers | 98 | 129265 | 67.5 (64.4–70.6) | 47 | 58906 | 21.7 (18.5–24.9) | 83 | 92347 | 19.8 (17.7–21.9) | 28 | 57789 | 54.1 (46.5–61.7) |
| Patients with chronic diseases | 52 | 166278 | 67.4 (63.9–70.9) | 29 | 152918 | 19.6 (16.9–22.2) | 40 | 160584 | 16.9 (13.9–19.9) | 19 | 132502 | 39.3 (31.9–46.7) |
| Pregnant/breastfeeding women | 13 | 25102 | 54.0 (46.3–61.7) | 3 | 2949 | 24.2 (19.4–28.9) | 12 | 9252 | 41.9 (33.0–50.8) | 3 | 2955 | 7.3 (1.7–12.8) |
| University students | 45 | 50471 | 67.7 (62.7–72.8) | 20 | 30469 | 19.6 (14.8–24.4) | 38 | 45910 | 20.7 (17.2–24.2) | 16 | 18937 | 43.7 (31.2–56.1) |
| Children and adolescents | 46 | 363825 | 70.7 (67.6–73.9) | 16 | 70757 | 25.2 (19.0–31.4) | 33 | 338814 | 19.8 (17.9–21.7) | 6 | 3969 | 37.9 (22.5–53.4) |
| Others | 58 | 164404 | 65.9 (60.8–71.0) | 35 | 103987 | 19.6 (16.4–22.8) | 52 | 152774 | 21.7 (18.4–25.1) | 23 | 86082 | 39.4 (26.1–52.8) |

For each pooled estimate, heterogeneity tests between studies reached Higgins' $I^2$ statistic > 92%, $P < 0.001$.

**Cross-country comparison of COVID-19 vaccination acceptance**. Among adults in the 58 studied countries, Brazil reported the highest acceptance (86.9%, 81.4–92.5), while Syria reported the lowest acceptance (35.9%, 34.3–37.5) (Fig. 2, Sheet 3 in Supplementary Data 2). Vaccination acceptance rates varied across the 58 countries: 12 countries exceeded 80% (Brazil, Indonesia, Nepal, Iran, Malaysia, mainland China, Argentina, UK, Canada, Italy, Spain, Paraguay); 17 countries were between 70% and 80% (Mexico, Vietnam, Australia, Somalia, Israel, Germany, France, New Zealand, Botswana, Netherlands, Bosnia and Herzegovina, India, USA, Pakistan, United Arab Emirates, Mozambique, Peru); 18 countries were between 60% and 70% (South Africa, Mongolia, Poland, Greece, Ukraine, Egypt, Ireland, Bangladesh, Palestine, Japan, Iraq, Philippines, Romania, Norway, Croatia, Turkey, Qatar, Korea); another 6 countries were between 50% and 60% (Hungary, Saudi Arabia, Democratic Republic of Congo, Nigeria, Kuwait, Ethiopia); and the remaining 5 countries were below 50% (Russia, Ghana, Jordan, Lebanon, Syria).

Among healthcare workers in the 47 studied countries, the acceptance rate of COVID-19 vaccination was the highest in Malaysia (97.7%, 95.8–99.7), and the lowest in Egypt (26.9%, 21.5–32.3). Vaccination acceptance rates also varied across the 47 countries: 12 countries exceeded 80% (Malaysia, Brazil, Thailand, Mexico, Nepal, Germany, Mongolia, mainland China, Kuwait, Italy, Romania, Poland); 11 countries were between 70% and 80% (Ghana, Israel. UK, USA, Vietnam, France, Canada, Australia, India, Qatar, Pakistan); 7 countries were between 60% and 70% (Belgium, Turkey, Saudi Arabia, Greece, Morocco, Spain, Sudan); 8 countries were between 50% and 60% (Palestine, South Africa, Tunisia, Lebanon, Malta, Ethiopia, Barbados, Jordan); and the remaining 9 countries were below 50% (Nigeria, Kenya, Japan, Togo, Switzerland, Oman, Cyprus, Democratic Republic of the Congo, Egypt).

Among patients with chronic diseases in the 23 studied countries, Japan reported the highest acceptance rate (89.8%, 85.5–94.2), while Turkey reported the lowest acceptance rate (29.2%, 25.9–32.5) (Fig. 3, Sheet 3 in Supplementary Data 2). Among pregnant or breastfeeding women, Belgium had the highest acceptance rate (78.8%, 77.7–79.8) among the 14 studied countries, whereas Qatar had the lowest (24.9%, 20.3–29.5). Among university students across the 23 studied countries, Japan had the highest acceptance rate (92.9%, 90.7–95.0), while Morocco had the lowest acceptance (27.0%, 24.5–29.4). Among children and adolescents in the 37 studied countries, Mexico reported the highest acceptance rate (95.2%, 95.0–95.4), while Turkey reported the lowest acceptance (36.3%, 33.4–39.3).

**Time trends of COVID-19 vaccination acceptance**. Figure 4 and Sheet 4 in Supplementary Data 2 display time trends for COVID-19 vaccination acceptance among adults and healthcare workers. Among adults, COVID-19 vaccination acceptance fell globally between February and December, 2020, then recovered during December 2020 to June 2021, and further dropped in late 2021. In the United States, the acceptance rate of COVID-19 vaccination first declined from 71.9% (95% CI 66.3-77.6) in April to 57.3% (53.2–61.5) in October, 2020, then increased to 88.1% (84.5–91.6) in June, 2021. The acceptance rate of COVID-19 vaccination across the United Kingdom dropped from 84.5% (73.1–95.8) to71.1% (69.5–72.8) during the period from April to October, 2020, and then increased to 97.7% (97.5–97.9) in February, 2021. In China, the acceptance rate of COVID-19 vaccination showed a decreasing trend from 91.3% (90.1–92.5) to 60.4% (57.3–63.4) from March to October, 2020, then recovered to 97.5% (97.3–97.8) in August, 2021, and further dropped in December, 2021. As for the remaining studied countries except the United

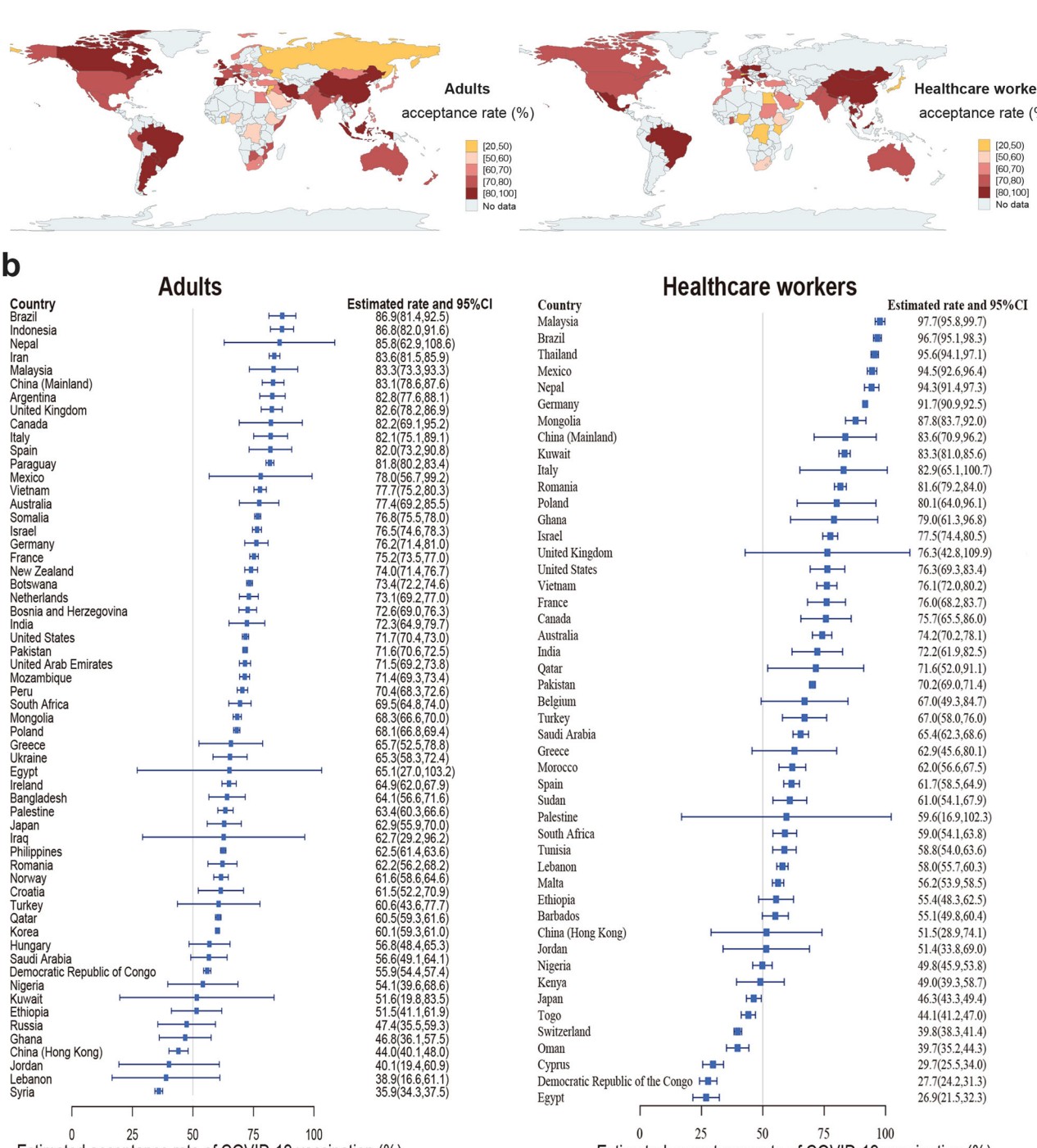

**Fig. 2 Estimated COVID-19 vaccination acceptance rates among adults and healthcare workers worldwide (%). a** Estimated COVID-19 vaccination acceptance rates among adults ($n = 6,538,325$) and healthcare workers ($n = 128,515$) across countries are shown via map. Different colors refer to different levels of acceptance rates. **b** Estimated COVID-19 vaccination acceptance rates and 95% confidence interval (CI) in adults ($n = 6,538,325$) and healthcare workers ($n = 128,515$) across countries are shown.

States, the United Kingdom, and China, it declined from 70.7% (66.0–75.4) to 47.6% (37.0–58.1) from April to December, 2020, then increased to 88.4% (82.0–94.8) in June, 2021, and further experienced some drops in the second half of 2021. In addition, time trends for COVID-19 vaccination acceptance among healthcare workers declined from 79.1% (68.9–89.4) to 53.7% (48.7–58.7) from May to September, 2020, then continuously increased to 85.8 (84.6–87.0) in August, 2021.

**Comparison of COVID-19 vaccination acceptance and uptake by sociodemographic subgroups.** Among adults, the COVID-19 vaccination acceptance rate of males (68.3%, 65.5–71.1) was higher than females (64.7%, 61.4–68.0) (Fig. 5, Sheet 5 in Supplementary Data 2). Those over 60-years-old (75.2%, 72.2–78.3) had the highest rate, following by those 40~59-years-old (68.2%, 64.8–71.7), and those 18~39-years-old (65.6%, 62.5–68.7). Asians (79.2%, 74.5–83.9) had the highest rate, while Black individuals

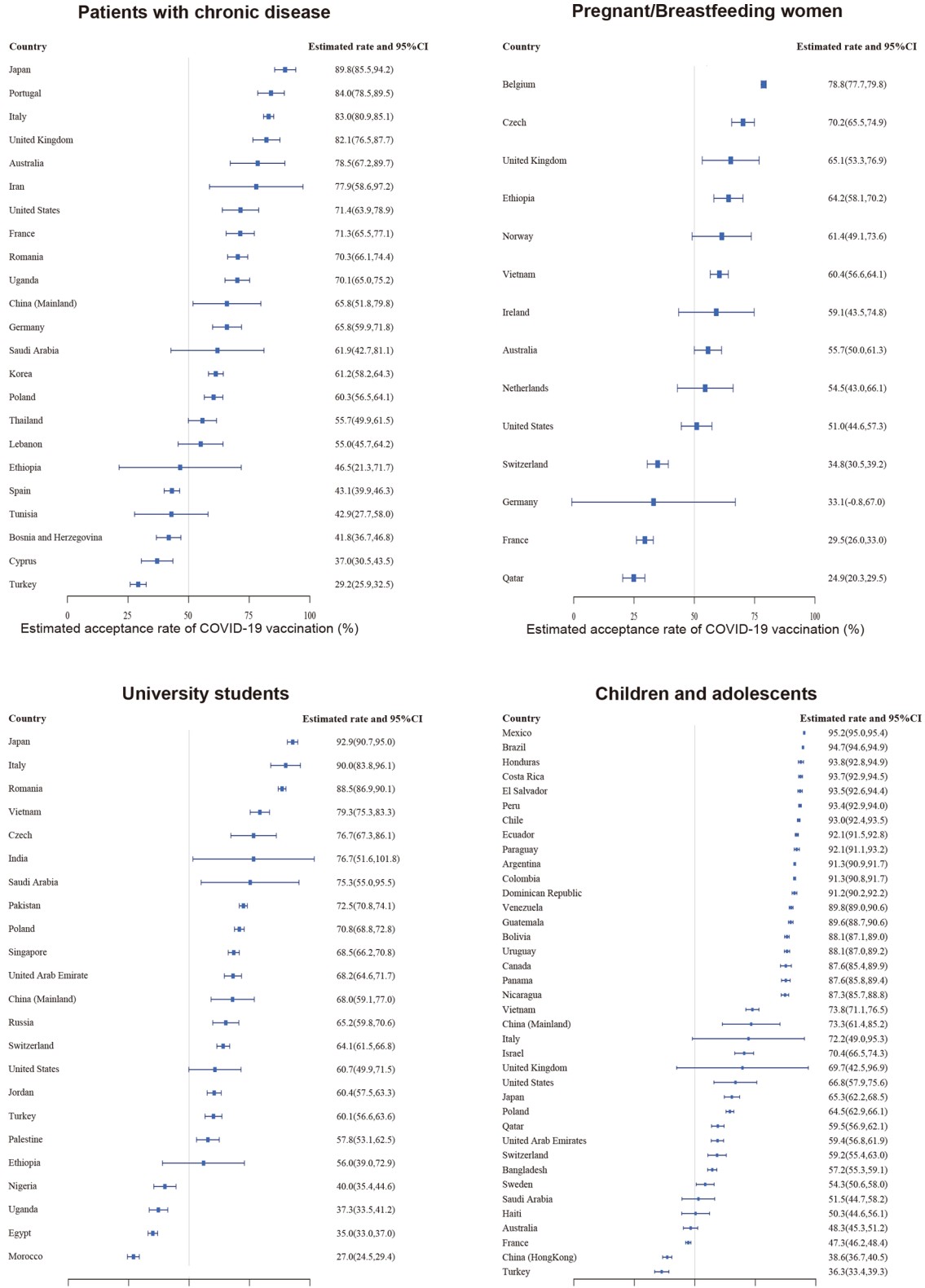

**Fig. 3 Estimated COVID-19 vaccination acceptance rates in different populations worldwide (%).** Estimated COVID-19 vaccination acceptance rates and 95% confidence interval (CI) in different populations across countries are shown. The populations include patients with chronic disease ($n = 165{,}438$), pregnant/breastfeeding women ($n = 25{,}102$), university students ($n = 43{,}832$), and children and adolescents ($n = 358{,}429$).

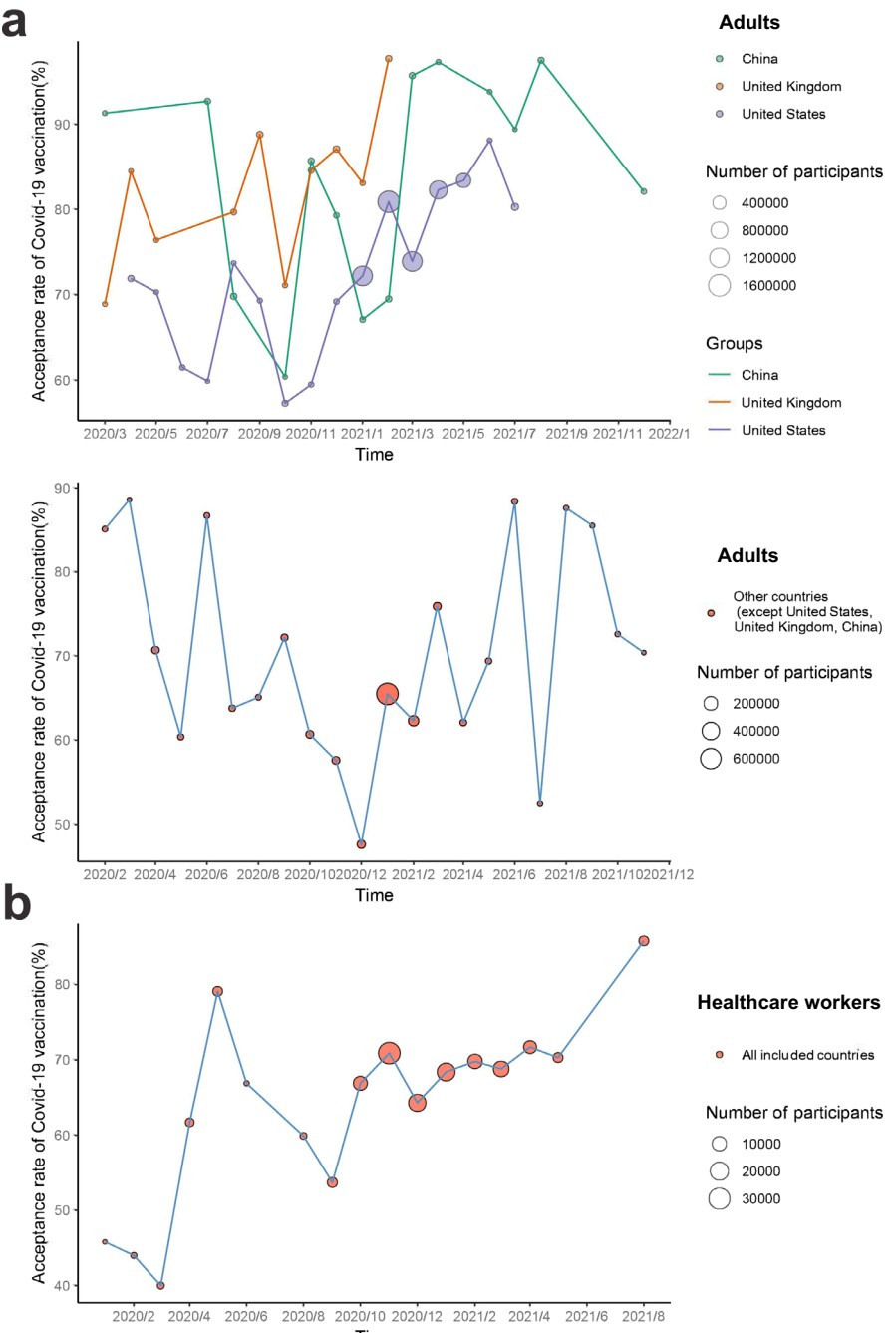

**Fig. 4 Time trends of COVID-19 vaccination acceptance rates between February 2020 and December 2021 among adults and healthcare workers.** **a** Time trends of COVID-19 vaccination acceptance for adults by country are shown in line graphs. Different colors refer to different countries, including China ($n = 358,429$ participants), the United Kingdom ($n = 105,593$ participants), the United States ($n = 5,816,222$ participants), and other countries ($n = 1,008,126$ participants). **b** Time trend for healthcare workers ($n = 128,090$) in all included countries is shown. Size of circles refers to number of study participants.

(49.1%, 41.6–56.6) had the lowest rate. Acceptance rates of those with a high school or below education, those with some college, and those with a bachelor's degree or higher were 61.7% (95% CI: 57.4–65.9), 64.5% (95%CI: 59.5–69.4), 68.7% (95%CI: 64.4–73.0) respectively. People with high incomes (74.2%, 70.4–78.0) had the highest rate, followed by those with middle incomes (70.5%, 65.8–75.3), and low incomes (64.8%, 59.9–69.7).

In addition, the uptake rate of COVID-19 vaccination for the Black population (59.2%, 40.5–77.8) was lower than the white and Asian by 6.5% and 19.4% respectively. The uptake rate for the

lowest-level groups of education or income (around 45%) was lower than the highest-level groups by around 20%.

## Discussion
The estimated acceptance rate of COVID-19 vaccination (67.8%) in our review was far low to achieve the threshold of herd immunity for the omicron variant[5,23]. Although reaching the threshold of herd immunity is only the most optimal situation, acceptance and uptake of COVID-19 vaccines need to be

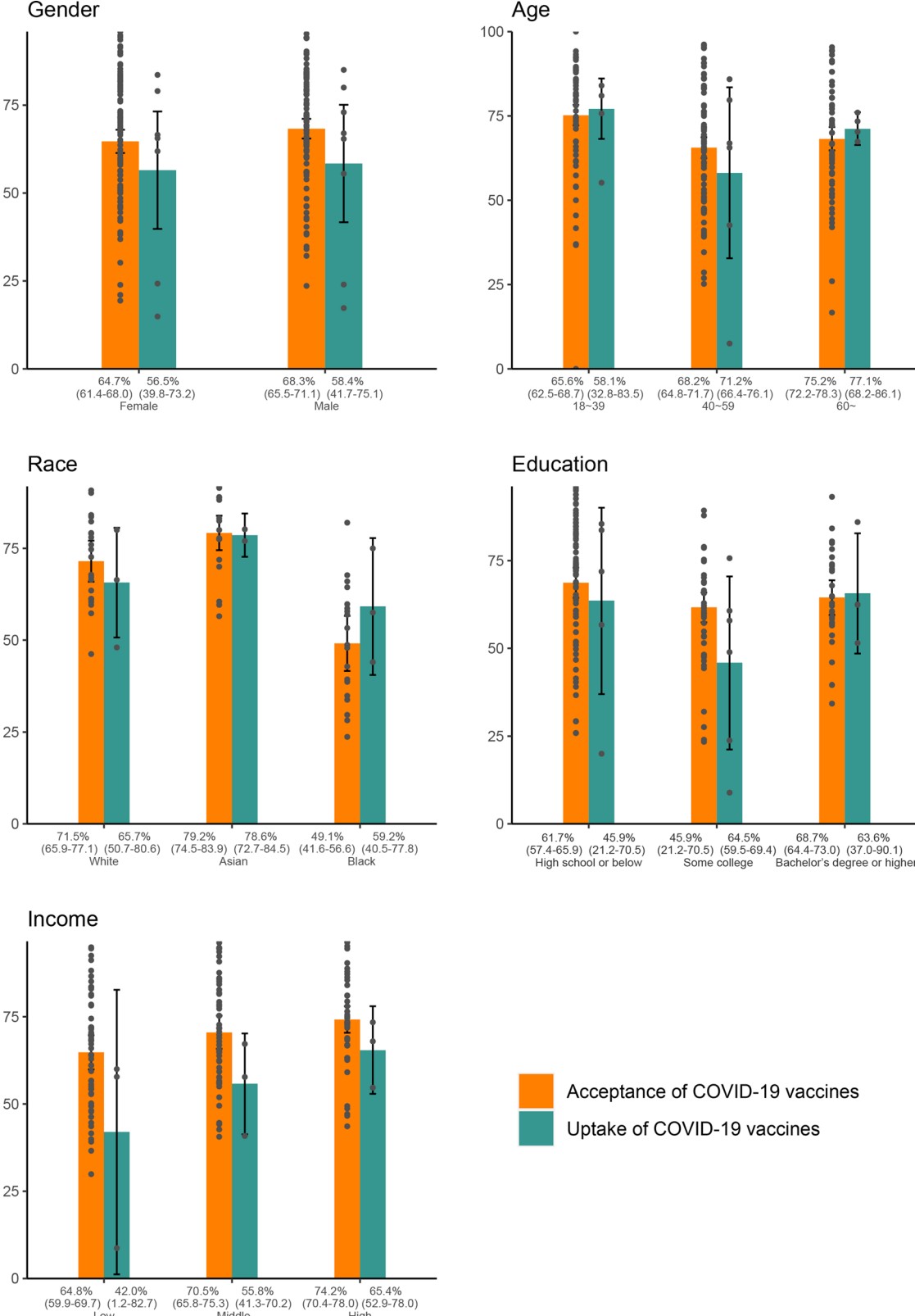

**Fig. 5 COVID-19 vaccination acceptance and uptake rates by subgroups for adults.** Estimated acceptance (orange block) and uptake (green block) rates and 95% confidence interval (CI) of COVID-19 vaccination by subgroups for adults are shown. The subgroups include gender ($n = 819,147$ participants), age ($n = 522,343$ participants), race ($n = 51,286$ participants), education ($n = 269,342$ participants), and income ($n = 298,195$ participants).

improved globally to protect people, especially for vulnerable groups such as patients with chronic diseases and pregnant/breastfeeding women. For most population groups, the acceptance rate was around 65–70%, with except for pregnant/breastfeeding women lowest at 54%. There seemed lack of safety data on COVID-19 vaccines for pregnancy and breastfeeding women[24], and this group was also more hesitant to other vaccination[25]. The vaccination uptake of pregnancy and breastfeeding women was also the lowest at 7.3%, far lower than the overall rate (42.3%).

In addition, vaccination acceptance varied by demographic and social-economic characteristics. Consistent with previous literature[19,26–29], females, those aged < 60 years old, Black individuals, lower educated persons, and lower income persons had lower acceptance of COVID-19 vaccination than their counterparts, and thus should be paid more attention. Populations with low socioeconomic status also faced the bigger obstacles to vaccination since their vaccine uptake rate was much lower than their acceptance, with around 20% gap between acceptance to uptake. To meet their acceptance, barriers to vaccine access need to be further addressed for the populations with low socio-economic status.

In this review, a large cross-country variation in COVID-19 vaccination acceptance rate was found for each population group, and countries with lower acceptance rates need more attention. The low acceptance rates (< 50%) of adults among six countries may be related to the widespread embrace of conspiracy beliefs and dissemination of misinformation about COVID-19, which subsequently causes negative attitudes towards vaccination[11,30]. High acceptance rates (> 80%) of adults among 12 countries may be associated with perceived risk of COVID-19 infection, trust in government, and stronger confidence in vaccine safety and effectiveness[12,23,31]. Reasons for not accepting COVID-19 vaccines should be investigated for each country, and targeted measures should be taken to improve COVID-19 vaccine acceptance according to their local contexts. And a transparent, reasonable, and robust immunization process can improve public confidence in the COVID-19 vaccine[31].

Globally, COVID-19 vaccination acceptance among adults showed first a declining trend from the beginning of the pandemic, and then an increasing trend starting at the end of 2020. Consistent with our findings, two systematic reviews[18,23] indicated the declining trend of COVID-19 vaccination acceptance before October 2020, and previous studies[32,33] also showed increasing trends in vaccine acceptance from September 2020 to February 2021. With the unprecedented speed of global research and development of COVID-19 vaccines, public concerns on vaccine safety and effectiveness may contribute to the declining acceptance of COVID-19 vaccines in the first year of the pandemic[19]. Misinformation about COVID-19 and distrust in governments could have also contributed to this declining trend in COVID-19 vaccination acceptance[34]. However, the recovery of vaccine acceptance starting from December 2020 was likely due to the threat of a second wave pandemic[35] and the reported high effectiveness of COVID-19 vaccines, such as Pfizer's announcement of over 90% vaccine effectiveness[36]. Moreover, many countries such as the United Kingdom and United States had officially recommended vaccination and started national vaccination campaigns since December 2020, which would apparently promote people's acceptance on COVID-19 vaccines[37,38].

However, vaccine acceptance of adults experienced some drops again in late 2021, although there was the continuous increase during 2021 for healthcare workers. As the virus mutates, reports of vaccine breakthrough cases and the calls for a booster dose of the vaccine may further speak to limited confidence in the vaccine effectiveness for protecting against COVID-19[39]. This reversal of the increasing trends in vaccine acceptance highlights the long-term need of health communication on COVID-19 vaccines in eliminating the long-term COVID-19 existence. To improve future vaccine acceptance globally, targeted communication and education campaigns should be strengthened and become a long-term action, and the supply of COVID-19 vaccines also needs to be ensured to improve accessibility.

Our study has several limitations. First, most studies applied convenience sampling due to the pandemic, and sample representability and selection bias need to be considered carefully. Second, since we cannot categorize continuous variables on vaccine acceptance consistently to achieve a meta-analysis, this review only included studies that did not use continuous measures of vaccine acceptance and may thus be biased. Third, although this review included more than 50 countries, it may not be representative of the global response to COVID-19 vaccine. In fact, data from low-income countries are still limited, and it is urgent to rapidly estimate vaccination acceptance in these countries to improve vaccine rollout worldwide. Fourth, vaccination acceptance is influenced by many factors, including conspiracy beliefs, vaccine confidence and other psychological factors, which may vary across different countries and times. However, these psychological factors are usually measured with different scales, which hindered the meta-analysis, and not the focus of this review. Therefore, we did not explore the influence of these psychological factors, and only performed subgroup analysis to deal with heterogeneity by sociodemographic characteristics. Finally, the data from the included studies on COVID-19 vaccination uptake is limited, which may not be representative of the global COVID-19 vaccination rate.

## Conclusion
In conclusion, COVID-19 vaccine acceptance needs to be improved globally. COVID-19 vaccination acceptance varied across different populations and countries, and changed over time. Continuous vaccine acceptance monitoring, especially for countries and sub-populations with low acceptance, is necessary to inform public health decision making.

## Data availability
All data generated or analysed in this study are included in this published article (and its supplementary files). Source data for the main figures in the manuscript are provided in Supplementary Data 2.

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

## Acknowledgements

This research was commissioned by the National Institute of Health Research using Official Development Assistance (ODA) funding, (Grant ID: 16/137/109). The views expressed in this publication are those of the author(s) and not necessarily those of the NHS, the National Institute for Health Research or the Department of Health. This research was supported by National Natural Science Foundation of China (No. 71874034), and the Soft Science Research Project of Shanghai "Science and Technology Innovation Action Plan" (22692107600). The funders had no role in study design, data collection and analysis, preparation of the paper, or the decision to publish. We thank Ying Zhang and Zehua Xu from School of Public Health, Fudan University for their help in data extraction. We also thank Yi Song from School of Public Health, Fudan University and Xudong Cui from School of Public Health, Lanzhou University for their help in figure formation.

## Author contributions

Z.H. conceptualized the study design. S.H. conducted literature searches. Q.W. and F.D. developed data extraction framework and selected the tool for quality assessment. F.D., Q.W., S.H., S.Z., Y.X., Z.Q., and X.Z. performed selecting the studies, data extraction, and quality assessment. Q.W. conducted meta-analyses. Qian Wang and Simeng Hu interpreted results and wrote the manuscript. Z.H. and L.L. critically revised the manuscript and contributed to the final version. All authors read and approved the version of the manuscript to be published.

## Competing interests

The authors declare no competing interests.
