## [Peer Review File · Communications Medicine]

Reviewers' comments:

Reviewer #1 (Remarks to the Author):

Inglese (US)

I read with interest the paper by Wang and colleagues. It is a review/metanalysis paper related to vaccine acceptance around the world.

The paper certainly deals with a topic of topical interest, although it is part of an area in which much has already been written. What is most interesting is the classification of the different age and occupational or risk classes.

I think the paper can be of interest to the scientific community even if I would have some issues to ask for an answer to the authors.

1. Has a sensitivity been done by adding the worst quality studies to the analysis? If yes, were there significant differences?
2. I do not see the funnel plots. Can the authors do these types of analyses and show them in the supplements?
3. Can a bibliography of the included studies also be made if journal rules allow?
4. I don't know if the inclusion of the words (survey related in the PICO has somehow over-specified the search). I point out, for example, knowing it very well, that the study available at the following link (<https://pubmed.ncbi.nlm.nih.gov/33924534/>) is not among those considered even though it is a survey of April 2021 and has a sample of 1000 people. I suggest the authors to verify the search strategy.

However, it must be said that every bit of information on the subject is useful even if, before publishing, I would like to ask the authors for some background information:

1. It is not clear to me how people were contacted to answer the questionnaire. Given that this is a convenience sample and selection bias is always around the corner (as the authors acknowledged in the limits) I wonder exactly how the people who responded were selected.
2. It is not at all clear to me why there is this pairing between Liguria and Switzerland.

Reviewers #2 and #3 (Remarks to the Author):

Review of "Mapping global acceptance..."

by Wang et al.

Reviewer: Stephan Lewandowsky and Philipp Schmid

Summary and Overall Recommendation

The authors aim to estimate the global acceptance of COVID-19 vaccination based on existing peer-

reviewed research articles and preprints. Providing aggregates of global acceptance rates can be highly relevant for scientists, practitioners, and policy makers. The goals of this paper are therefore commendable.

However, the usefulness of such aggregates depends heavily on the timing of the measurement. In addition, the selection method of the individual papers is central to the reduction of bias in the aggregate. These two points constitute major shortcomings of the paper that prevent us from endorsing publication at this juncture.

1) The systematic review covers publications from February 2020 to April 2021 and compares the acceptance rates with herd immunity thresholds. Considered from the perspective of early 2022, this timeframe is too limited to inform policy makers about current levels of herd immunity levels. Current national data on uptake rates would provide a more up-to-date indicator of whether herd immunity will be reached in 2022. We realize that the need to be current never ends; however, being 9 months out of date in a fast-moving environment is too much.

2) An advantage of a systematic review is its scientific rigor. However, that rigor may be compromised by the authors' decision to exclude studies that "applied continuous variables to evaluate vaccination acceptance". This decision is not further justified and provides a potential source for serious bias. The social sciences have developed several validated measures of vaccine hesitancy in recent years. Almost all of these validated measures consist of continuous variables. Thus, it can be expected that much of the most relevant research to assess hesitancy is not covered in this meta-analysis. It could therefore be argued that the authors do not focus on hesitancy but rather on acceptance (this difference is not discussed by the authors; see points below). But even then, the problem remains. Many, if not most social scientific studies on vaccine acceptance use measures of intention that consist of continuous scales. Excluding these studies will most likely lead to biased estimates.

In addition to these principal problems, a revision (if it is invited) should address the following major points.

Major points

- The introduction lacks a clear definition of vaccine hesitancy, especially in distinction to acceptance. These phenomena are not synonyms but are used as such in the text.
- The search string used by the authors is rather limited. The authors chose the survey terms (survey OR questionnaire OR poll) as a third block for the search string. This branch does not cover studies that simply do not mention these terms but rather use, for example, the term "study".
- Some relevant information is missing on the general inclusion/exclusion strategy. Did the authors only searched title and abstract if articles when using the search string? Did the authors use quantitative reliability assessments of raters?
- It is unclear whether the individual papers included vaccinated individuals in their assessment of acceptance rates. It is obvious, for example, that acceptance rates decline when vaccinated individuals are not included in the assessment. The more individuals get vaccinated the higher the proportion of unvaccinated individuals who are also not willing to ever get the vaccine. How was uptake accounted for in the single studies and the review? This is especially relevant when drawing conclusions for herd immunity.
- The authors try to explain differences in acceptance rates in the discussion by listing interesting potential predictors (e.g., embrace of conspiracy beliefs, confidence in vaccinations). However, these are simply assumptions without presenting insights from the reviewed papers. It would be an interesting contribution to compare/discuss predictors of acceptance rates rather than acceptance alone. Predictors can inform practitioners and policy makers on how to design tailored interventions

and could contribute to the authors' own recommendation "To improve future vaccine acceptance globally, targeted communication and education campaigns should be strengthened to convince the public of the safety and effectiveness of COVID-19 vaccines".

- The authors explain the increase of vaccine acceptance after November 2020 with the increased threat of a second wave and results from scientific studies about high effectiveness. But what about official recommendations? Many countries started to recommend the vaccine only in December 2020 or even later. Recommendations from NITAGS and other official committees are most likely highly relevant for acceptance rates and those factors should be entered into the analysis. It is known that vaccination campaigns generate their own momentum once they commence.
- The Discussion was too verbose and repeated much information that was made available earlier (e.g., in the Results). It should be cut by 50%.

Detailed comments

[line#]

27-28 Why not also Google Scholar? This requires a brief explanation.

67 Three-quarters seems to be an underestimate given Omicron.

149 What does "resident or adult" mean? Why not just "adults"?

164-165 How exactly were the 501 other articles excluded? Some more explanation would be helpful here.

Response to Reviewers' comments:

Reviewer #1 (Remarks to the Author):

Inglese (US)

I read with interest the paper by Wang and colleagues. It is a review/metanalysis paper related to vaccine acceptance around the world.

The paper certainly deals with a topic of topical interest, although it is part of an area in which much has already been written. What is most interesting is the classification of the different age and occupational or risk classes.

I think the paper can be of interest to the scientific community even if I would have some issues to ask for an answer to the authors.

1. Has a sensitivity been done by adding the worst quality studies to the analysis? If yes, were there significant differences?

Response: In the revised version, we excluded pre-print articles and the studies that scored below 13 points in quality assessment to avoid potential bias caused by low-quality studies. Instead, we did a sensitivity analysis with studies whose sample size was more than 300, and got similar results (*Additional file 1 Table S4*).

In the methods section (page 9, lines 163-164), we added “*Additionally, we did a sensitivity analysis of the pooled acceptance rate of COVID-19 vaccination with studies whose sample size was more than 300 (Additional file 1 Table S4).*”

In addition, we had updated the included studies to early 2022. Now a total of 519 articles are included in our review.

2. I do not see the funnel plots. Can the authors do these types of analyses and show them in the supplements?

Response: We used the heterogeneity tests with Higgins' I^2 statistic to assess variability between studies (page 9, line 146). We found that heterogeneity between acceptance estimates of COVID-19 vaccination is high ($I^2 > 75\%$), and due to the high heterogeneity among these studies, we did not use funnel plots to assess publication bias.

3. Can a bibliography of the included studies also be made if journal rules allow?

Response: We now added a bibliography of the included studies in appendix (*Additional file 1*).

4. I don't know if the inclusion of the words (survey related in the PICO has somehow over-specified the search). I point out, for example, knowing it very well, that the study available at the following link (<https://pubmed.ncbi.nlm.nih.gov/33924534/>) is not among those considered even though it is a survey of April 2021 and has a sample of 1000 people. I suggest the authors to verify the search strategy.

Response: In our inclusion and exclusion criteria, we excluded studies which assessed willingness-to-pay or conditional acceptance (page 6, lines 98-100). In the study you mentioned (<https://pubmed.ncbi.nlm.nih.gov/33924534/>), the question “A

COVID-19 vaccine has already been approved. If you were offered to get the vaccine in the next months at no cost for you, how likely are you to take it?" limited the time to get vaccination. We categorized this study as conditional acceptance, and excluded this study.

Reviewer: However, it must be said that every bit of information on the subject is useful even if, before publishing, I would like to ask the authors for some background information:

5. It is not clear to me how people were contacted to answer the questionnaire. Given that this is a convenience sample and selection bias is always around the corner (as the authors acknowledged in the limits) I wonder exactly how the people who responded were selected.

Response: Due to the COVID-19 epidemic, many countries have taken strict restrictions to reduce the disease spread, such as lockdown or quarantine. Considering this tough environment, most studies distributed the questionnaire through social media or Internet. We described the details of sampling methods of each included study in *Additional file 2 Sheet 1* (this is an excel sheet since many extraction data on article description, acceptance and uptake are shown for each study).

6. It is not at all clear to me why there is this pairing between Liguria and Switzerland.

Response: In this study, we first categorized all study participants into seven population groups, and within each population group, we then estimated the acceptance of COVID-19 vaccination in individual countries by synthesizing all studies from the same country. And we compared the acceptance rates of COVID-19 vaccination across these countries.

Reviewers #2 and #3 (Remarks to the Author):
Review of “Mapping global acceptance...” by Wang et al.
Reviewer: Stephan Lewandowsky and Philipp Schmid

Summary and Overall Recommendation

The authors aim to estimate the global acceptance of COVID-19 vaccination based on existing peer-reviewed research articles and preprints. Providing aggregates of global acceptance rates can be highly relevant for scientists, practitioners, and policy makers. The goals of this paper are therefore commendable.

However, the usefulness of such aggregates depends heavily on the timing of the measurement. In addition, the selection method of the individual papers is central to the reduction of bias in the aggregate. These two points constitute major shortcomings of the paper that prevent us from endorsing publication at this juncture.

1. The systematic review covers publications from February 2020 to April 2021 and compares the acceptance rates with herd immunity thresholds. Considered from the perspective of early 2022, this timeframe is too limited to inform policy makers about current levels of herd immunity levels. Current national data on uptake rates would provide a more up-to-date indicator of whether herd immunity will be reached in 2022. We realize that the need to be current never ends; however, being 9 months out of date in a fast-moving environment is too much.

Response: We agree that any research on COVID-19 should be time-sensitive to help policy makers. Therefore, we added 393 peer-reviewed articles that published from June 24, 2021 to February 27, 2022 to update the pooling of included studies. Now a total of 519 articles are included in our review. All the included articles in our study can be find in appendix (*Additional file 2 Sheet 1*: this is an excel sheet since many extraction data on article description, acceptance and uptake are shown for each study).

And national data on uptake rates can be a source for policy-maker, but lack the variation across populations. In the revised version, we added uptake rates by different populations and demographic or socioeconomic characteristics (*Table 1 and Figure 4*).

2. An advantage of a systematic review is its scientific rigor. However, that rigor may be compromised by the authors’ decision to exclude studies that “applied continuous variables to evaluate vaccination acceptance”. This decision is not further justified and provides a potential source for serious bias. The social sciences have developed several validated measures of vaccine hesitancy in recent years. Almost all of these validated measures consist of continuous variables. Thus, it can be expected that much of the most relevant research to assess hesitancy is not covered in this meta-analysis. It could therefore be argued that the authors do not focus on hesitancy but rather on acceptance (this difference is not discussed by the authors; see points below). But even then, the problem remains. Many, if not most social scientific studies on vaccine acceptance use measures of intention that consist of continuous scales. Excluding these studies will most likely lead to biased estimates.

Response: Sorry that these two terms “acceptance” and “hesitancy” led to confusion. Our review focused on vaccine acceptance, and therefore in the introduction section, we deleted all contents relating to vaccine hesitancy and used vaccine acceptance instead.

In addition, vaccine acceptance rate is our key measure in the meta-analysis, which is calculated using category variables and cannot be achieved by continuous variables. Studies with continuous variables usually used different measurement scales, and the results cannot be easily compared across measurement scales. In our selection process of studies, we also found that most studies used category variables to assess vaccine acceptance, with a few studies using continuous scales. Therefore, to conduct meta-analysis, we excluded studies that “applied continuous variables to evaluate vaccination acceptance”.

In our review, 519 articles were finally eligible for inclusion. The number of included studies is huge and much larger than most systematic reviews, which minimized the estimation bias.

We added the following statements in page 6-7, lines 101-103:

“Studies using continuous variables usually adopted different measurement scales with incomparable findings, and therefore were excluded in our review for conveniently conducting the meta-analysis of vaccination acceptance rate.”

Reviewer: In addition to these principal problems, a revision (if it is invited) should address the following major points.

Major points

3. The introduction lacks a clear definition of vaccine hesitancy, especially in distinction to acceptance. These phenomena are not synonyms but are used as such in the text.

Response: Sorry that these two terms led to confusion. Our review focused on vaccine acceptance, and therefore we deleted all contents relating to vaccine hesitancy and used vaccine acceptance instead.

4. The search string used by the authors is rather limited. The authors chose the survey terms (survey OR questionnaire OR poll) as a third block for the search string. This branch does not cover studies that simply do not mention these terms but rather use, for example, the term “study”.

Response: There are many social media studies on COVID-19 vaccine attitudes and acceptance, which are not our target studies due to their skewed biased samples. Therefore, we limit the study design of target studies to surveys only. In this review, we used the search terms (survey OR questionnaire OR poll) in [all fields] instead of [title or abstract] only, which can cover all survey studies. And we found that this search terms were also used in previous systematic reviews relating to COVID-19 pandemic.

5. Some relevant information is missing on the general inclusion/exclusion strategy. Did the authors only searched title and abstract if articles when using the search string? Did the authors use quantitative reliability assessments of raters?

Response: No, we searched [all fields] of articles instead of [title or abstract] only when using the search string. Please find in (*Additional file 1: Table S1. Search strategy*). And we also clearly stated this point in the methods section.

To ensure the reliability of raters' assessments, 1) for article selection, “*Two independent researchers (SH, QW) first screened titles and abstracts, and then scrutinized the full texts to estimate their eligibility. When they had disagreements on study selection, the third researcher (FD) was consulted.*” (page 7, lines 106-108)

2) for article extraction and quality assessment, “*Two researchers (SH, QW) independently performed article extraction and assessed the quality of included studies. When inconsistency arose, they were asked to discuss and revisit the article until reaching a consensus.*” (page 7, lines 112-114)

6. It is unclear whether the individual papers included vaccinated individuals in their assessment of acceptance rates. It is obvious, for example, that acceptance rates decline when vaccinated individuals are not included in the assessment. The more individuals get vaccinated the higher the proportion of unvaccinated individuals who are also not willing to ever get the vaccine. How was uptake accounted for in the single studies and the review? This is especially relevant when drawing conclusions for herd immunity.

Response: Generally, single studies on vaccine acceptance did not exclude those vaccinated individuals, and grouped them as “accept” group when calculating vaccine acceptance rate. In our review, we also done like this, taking those vaccinated individuals as “accept” group when calculating vaccine acceptance rate.

We added this in page 8, lines 136-138:

“For studies covering the vaccinated individuals, we took vaccinated individuals as the “accept” group when calculating vaccine acceptance rates.”

7. The authors try to explain differences in acceptance rates in the discussion by listing interesting potential predictors (e.g., embrace of conspiracy beliefs, confidence in vaccinations). However, these are simply assumptions without presenting insights from the reviewed papers. It would be an interesting contribution to compare/discuss predictors of acceptance rates rather than acceptance alone. Predictors can inform practitioners and policy makers on how to design tailored interventions and could contribute to the authors' own recommendation “To improve future vaccine acceptance globally, targeted communication and education campaigns should be strengthened to convince the public of the safety and effectiveness of COVID-19 vaccines”.

Response: We agree that predictors of acceptance rates are an interesting contribution and can inform how to design tailored interventions. However, predictors such as embrace of conspiracy beliefs and confidence in vaccinations are relatively subjective on measurements, and measurements are usually inconsistent across studies, which hindered the review and meta-analysis. Additionally, the influence of conspiracy and vaccine confidence on vaccine acceptance have been well studied/confirmed and out of our review. Therefore, our review focused on objective predictors such as population groups and sociodemographic characteristics.

We now added this limitation in page 17, lines 323-326:

“Third, vaccination acceptance is influenced by many factors, including conspiracy beliefs, vaccine confidence and other psychological factors, which may vary across different countries and times. However, these psychological factors are relatively subjective on measurements, and measurements are usually inconsistent across studies, which hindered the review and meta-analysis. Therefore, we did not explore the influence of these psychological factors, and only performed subgroup analysis to deal with heterogeneity by sociodemographic characteristics.”

8. The authors explain the increase of vaccine acceptance after November 2020 with the increased threat of a second wave and results from scientific studies about high effectiveness. But what about official recommendations? Many countries started to recommend the vaccine only in December 2020 or even later. Recommendations from NITAGS and other official committees are most likely highly relevant for acceptance rates and those factors should be entered into the analysis. It is known that vaccination campaigns generate their own momentum once they commence.

Response: Thanks for your suggestions. We now added this discussion point and made the corresponding changes following your suggestions (page 18, lines 306-308):

“Moreover, many countries such as the United Kingdom and United States had officially recommended vaccination and started national vaccination campaigns since December 2020, which would apparently promote people’s acceptance on COVID-19 vaccines (37-38).”

We added two articles as our reference 37 and 38:

37. Mathieu E, Ritchie H, Ortiz-Ospina E, Roser M, Hasell J, Appel C, et al. A global database of COVID-19 vaccinations. *Nature Human Behaviour*. 2021;5(7):947-53.

38. Abinaya E, Kumar MR, Ruckmani A, Arunkumar R. Assessment of COVID-19 Vaccine Acceptance among Health Care Workers and General Population-A Cross-Sectional Survey. *Journal of Communicable Diseases*. 2021;53(2):82-8.

9. The Discussion was too verbose and repeated much information that was made available earlier (e.g., in the Results). It should be cut by 50%.

Response: We have made the corresponding changes following your suggestions.

Detailed comments

10. [line#] 27-28 Why not also Google Scholar? This requires a brief explanation.

Response: Google Scholar is certainly a good source for article searching. But it includes not only peer-reviewed papers but also grey literature. Since there are huge studies on COVID-19 vaccine acceptance, we only focused on studies with high-quality. We used the popular publication databases and did not include grey literature. The databases we used have covered most published studies with high-quality, and therefore we did not use Google Scholar.

11. [line#] 67 Three-quarters seems to be an underestimate given Omicron.

Response 10: We made the corresponding changes as follow (page 5, lines 62-65):

“Considering the delta variant, around 85% of the population should get immunity through natural infection or vaccination (4). Given the powerful capability of the omicron variant to escape neutralizing antibodies elicited by current vaccines, more than 85% of the population need to get immunity (5).”

We have added two articles as our reference 4 and 5:

4. Burki TK. Omicron variant and booster COVID-19 vaccines. *The Lancet Respiratory Medicine*. 2022;10(2):e17.

5. Lu L, Mok BW-Y, Chen L-L, Chan JM-C, Tsang OT-Y, Lam BH-S, et al. Neutralization of Severe Acute Respiratory Syndrome Coronavirus 2 Omicron Variant by Sera From BNT162b2 or CoronaVac Vaccine Recipients. *Clin Infect Dis*. 2021;ciab1041.

12. [line#] 149 What does “resident or adult” mean? Why not just “adults”?

Response: Both “residents aged 18 years and above” and “adults” were from included studies. To avoid misunderstanding, we have changed “resident or adult” into “adults”.

13. [line#] 164-165 How exactly were the 501 other articles excluded? Some more explanation would be helpful here.

Response: Figure 1 showed the details of the excluded studies. In the revised version, 974 articles were excluded following the full-text articles assessment for eligibility. Among these excluded articles, (1) 21 were duplicate studies and data; (2) 7 completed data analysis through indirect methods; (3) 6 studied on clinical-trial, emergency or boosting vaccination acceptance; (4) 59 lack of results on outcomes of interest; (5) 142 scored below 13 points in quality assessment; (6) 739 included conditional inquiries.

REVIEWERS' COMMENTS:

Reviewer #1 (Remarks to the Author):

The authors have now answered to all my issues raised.
I would really want to thank them for the great job done.

Reviewers #2 and #3 (Remarks to the Author):

Review of "Mapping global acceptance..."

by Wang et al.

Reviewer: Stephan Lewandowsky and Philipp Schmid

Summary and Overall Recommendation

The authors aim to estimate the global acceptance of COVID-19 vaccination based on existing peer-reviewed research articles and preprints. Providing aggregates of global acceptance rates can be highly relevant for scientists, practitioners, and policy makers. The goals of this paper are therefore commendable.

We reviewed the paper at the first round and found it unsuitable for publication for several reasons.

First, we argued at the previous round that the initial timeframe of the review was not informative enough and that current national data on uptake rates provide a more up-to-date indicator of whether herd immunity will be reached. In response, the authors added 393 peer-reviewed articles published from June 24, 2021 to February 27, 2022 to update the pooling of included studies. In many cases national data is still more up to date and peer reviewed articles cannot compete with the speed with which these data are published. We appreciate the added subgroup analyses as they provide additional value that is not usually covered by national data. The coverage and acceptance rates from survey data within the new time frame provide interesting information even if they are not the most up-to-date database.

In response to our second concern regarding the exclusion of continuous measures of acceptance the authors responded, "In addition, vaccine acceptance rate is our key measure in the meta-analysis, which is calculated using category variables and cannot be achieved by continuous variables." And now state: "Studies using continuous variables usually adopted different measurement scales with incomparable findings, and therefore were excluded in our review for conveniently conducting the meta-analysis of vaccination acceptance rate." It is not entirely clear why these scales were classified as incomparable nor why a meta-analysis cannot be achieved by continuous variables. Several meta-analyses in different fields report average effects across studies that used different continuous scales (after standardizing the findings by rescaling responses to, say, 0-1). A meta-analysis on continuous outcomes, in addition to the meta-analysis of categorical outcomes, could reveal whether results depend on the way acceptance was measured or whether the authors conclusions are robust across measures. However, we acknowledge that this might be a time-consuming additional task. Thus, we recommend that the authors should at least mention in

the limitation section that their definition of acceptance only included studies that did not use continuous measures and may thus be biased. Furthermore, the argument about “incomparable findings” should be explained.

The authors deleted any mentioning of the term vaccine hesitancy and now only use “acceptance”, which is less confusing. However, we still recommend defining the term acceptance at least briefly in the introduction. A short definition will make it easier for readers to understand the difference to uptake and the selection of measures (intention etc.) Moreover, a definition may also explain why acceptance is used as a term in the article, but the references are about vaccine hesitancy? For example:

“Vaccine acceptance is a complex and context specific issue that varies across time, place, and vaccines (14).”

In response to the comment on psychological predictors the authors now state “However, these psychological factors are relatively subjective on measurements,...”. The meaning of this evaluation is rather unclear. Are psychological predictors not analysed because the authors considered them subjective? Because they were measured with different scales? Or because this was simply not the focus of the study?

We recommend further revision to address these points, although further review is unlikely to be required.

Response to REVIEWERS' COMMENTS:

Reviewer #1 (Remarks to the Author):

The authors have now answered to all my issues raised. I would really want to thank them for the great job done.

Response: Thanks.

Reviewers #2 and #3 (Remarks to the Author):

Review of “Mapping global acceptance...” by Wang et al.

Reviewer: Stephan Lewandowsky and Philipp Schmid

Summary and Overall Recommendation

The authors aim to estimate the global acceptance of COVID-19 vaccination based on existing peer-reviewed research articles and preprints. Providing aggregates of global acceptance rates can be highly relevant for scientists, practitioners, and policy makers. The goals of this paper are therefore commendable.

We reviewed the paper at the first round and found it unsuitable for publication for several reasons.

First, we argued at the previous round that the initial timeframe of the review was not informative enough and that current national data on uptake rates provide a more up-to-date indicator of whether herd immunity will be reached. In response, the authors added 393 peer-reviewed articles published from June 24, 2021 to February 27, 2022 to update the pooling of included studies. In many cases national data is still more up to date and peer reviewed articles cannot compete with the speed with which these data are published. We appreciate the added subgroup analyses as they provide additional value that is not usually covered by national data. The coverage and acceptance rates from survey data within the new time frame provide interesting information even if they are not the most up-to-date database.

Response: Thanks.

In response to our second concern regarding the exclusion of continuous measures of acceptance the authors responded, “In addition, vaccine acceptance rate is our key measure in the meta-analysis, which is calculated using category variables and cannot be achieved by continuous variables.” And now state: “Studies using continuous variables usually adopted different measurement scales with incomparable findings, and therefore were excluded in our review for conveniently conducting the meta-analysis of vaccination acceptance rate.” It is not entirely clear why these scales were classified as incomparable nor why a meta-analysis cannot be achieved by continuous variables. Several meta-analyses in different fields report average effects across studies that used different continuous scales (after standardizing the findings by rescaling responses to, say, 0-1). A meta-analysis on continuous outcomes, in addition to the meta-analysis of categorical outcomes, could reveal whether results depend on the way acceptance was measured or whether the authors conclusions are robust

across measures. However, we acknowledge that this might be a time-consuming additional task. Thus, we recommend that the authors should at least mention in the limitation section that their definition of acceptance only included studies that did not use continuous measures and may thus be biased. Furthermore, the argument about “incomparable findings” should be explained.

Response: In the limitation section, we now added a statement and made the corresponding changes following your suggestions (page 17, line 322-325):

“Second, since we cannot categorize continuous variables on vaccine acceptance consistently to achieve a meta-analysis, this review only included studies that did not use continuous measures of vaccine acceptance and may thus be biased.”

To explain the argument about “incomparable findings” more clearly, we made the corresponding changes (page 7, lines 101-104):

“Studies using continuous variables usually adopted different response ranges with no consistent meanings for response options across studies, and there were also no clear cut-off points for vaccination acceptance or refusal in continuous variables. Therefore, studies with continuous variables were excluded in our review for conducting the meta-analysis of vaccination acceptance rate.”

The authors deleted any mentioning of the term vaccine hesitancy and now only use “acceptance”, which is less confusing. However, we still recommend defining the term acceptance at least briefly in the introduction. A short definition will make it easier for readers to understand the difference to uptake and the selection of measures (intention etc.) Moreover, a definition may also explain why acceptance is used as a term in the article, but the references are about vaccine hesitancy? For example: “Vaccine acceptance is a complex and context specific issue that varies across time, place, and vaccines (14).”

Response: We now added the definition of vaccine acceptance in the introduction in page 5, lines 71-73:

“Vaccine acceptance is defined as the individual or group decision to accept or refuse, when presented with an opportunity to vaccinate (14). It is a complex and context specific issue that varies across time, place, and vaccines (15).”

In response to the comment on psychological predictors the authors now state “However, these psychological factors are relatively subjective on measurements...”. The meaning of this evaluation is rather unclear. Are psychological predictors not analysed because the authors considered them subjective? Because they were measured with different scales? Or because this was simply not the focus of the study?

Response: We now made the statement more clearly in page 17, lines 330-331:

“However, these psychological factors are usually measured with different scales, which hindered the meta-analysis, and not the focus of this review.”

We recommend further revision to address these points, although further review is unlikely to be required.

Response: Thanks. We have addressed all these points in this revision.